# Associations of psychosocial factors and cardiovascular health measured by Life's Essential 8: The Atherosclerosis Risk in Communities (ARIC) study

**Kennedy M. Peter-Marske**[1] *, **Anna Kucharska-Newton**[1], **Eugenia Wong**[1], **Yejin Mok**[2,3], **Priya Palta**[4], **Pamela L. Lutsey**[5], **Wayne Rosamond**[1]

1 Department of Epidemiology, Gillings School of Global Public Health, University of North Carolina at Chapel Hill, Chapel Hill, North Carolina, United States of America, 2 Department of Epidemiology, Johns Hopkins Bloomberg School of Public Health, Baltimore, Maryland, United States of America, 3 Welch Center for Prevention, Epidemiology and Clinical Research, Baltimore, Maryland, United States of America, 4 Department of Neurology, University of North Carolina at Chapel Hill, Chapel Hill, North Carolina, United States of America, 5 Division of Epidemiology & Community Health, School of Public Health, University of Minnesota, Minneapolis, Minnesota, United States of America

* kennedy_peter@unc.edu

**Data Availability Statement:** Data can be requested through the Biologic Specimen and Data

## Abstract

### Aims

Few studies investigate whether psychosocial factors (social isolation, social support, trait anger, and depressive symptoms) are associated with cardiovascular health, and none with the American Heart Association's new definition of cardiovascular health, Life's Essential 8 (LE8). Therefore, we assessed the cross-sectional associations of psychosocial factors with Life's Essential 8 and individual components of Life's Essential 8.

### Methods

We included 11,311 Atherosclerosis Risk in Communities cohort participants (58% females; 23% Black; mean age 57 (standard deviation: 6) years) who attended Visit 2 (1990–1992) in this secondary data analysis using cross-sectional data from the ARIC cohort study. Life's Essential 8 components included diet, physical activity, nicotine exposure, sleep quality, body mass index, blood lipids, blood glucose, and blood pressure. Life's Essential 8 was scored per the American Heart Association definition (0–100 range); higher scores indicate better cardiovascular health. Associations of categories (high, moderate, and low) of each psychosocial factor with continuous Life's Essential 8 score and individual Life's Essential 8 components were assessed using multivariable linear regressions.

### Results

11% of participants had high Life's Essential 8 scores (80–100), while 67% and 22% had moderate (50–79) and low Life's Essential 8 scores (0–49) respectively. Poor scores on psychosocial factor assessments were associated with lower Life's Essential 8 scores, with the

Repository Information Coordinating Center (BioLINCC) website (https://biolincc.nhlbi.nih.gov/studies/aric/) after creating an account and registering with the site. The data dictionary is available on this website. More information about the ARIC study can be found at https://aric.cscc.unc.edu/aric9/.

**Funding:** The Atherosclerosis Risk in Communities study has been funded in whole or in part with Federal funds from the National Heart, Lung, and Blood Institute, National Institutes of Health, Department of Health and Human Services, under Contract nos. (75N92022D00001, 75N92022D00002, 75N92022D00003, 75N92022D00004, 75N92022D00005). HbA1c data collection was supported by NIH/NIDDK grant R21 DK080294. The ARIC portion of the Sleep Heart Health Study was supported by National Heart, Lung, and Blood Institute cooperative agreements U01HL53934 (University of Minnesota) and U01HL64360 (Johns Hopkins University). Kennedy M. Peter-Marske was supported by an National Institutes of Health (NIH)/National Heart, Lung, and Blood Institute National Research Service Award (T32-HL007055). Pamela L. Lutsey was supported by NIH/NHLBII K24 HL159246. The funders had no role in the study design, data collection and analysis, decision to publish, or preparation of the manuscript.

**Competing interests:** The authors have declared that no competing interests exist.

largest magnitude of association for categories of depressive symptoms (low β = Ref.; moderate β = -3.1, (95% confidence interval: -3.7, -2.5; high β = -8.2 (95% confidence interval: -8.8, -7.5)). Most psychosocial factors were associated with Life's Essential 8 scores for diet, physical activity, nicotine, and sleep, but psychosocial factors were not associated with body mass index, blood lipids, blood glucose, or blood pressure.

## Conclusion

Less favorable measures of psychosocial health were associated with lower Life's Essential 8 scores compared better measures of psychosocial health among middle-aged males and females.

## Introduction

Psychosocial factors such as social isolation, social support, trait anger, and depressive symptoms, have been associated with the incidence of a variety of cardiovascular diseases (CVDs) such as myocardial infarction and stroke, CVD mortality, and the progression of CVD [1–5]. The majority of past studies have focused on the relationships these four psychosocial factors with clinically recognized CVD events [1,2,4]. However, little is known about how they are related to cardiovascular health (CVH) as defined by the American Heart Association (AHA). In 2010, the AHA set a goal of improving CVH in all Americans, defining CVH as Life's Simple 7 (LS7) [6], expanding the definition of CVH to include a spectrum of health, not limited to non-diseased and diseased states. CVH metrics created further opportunities for primary prevention of CVD through early risk identification and intervention. The LS7 metric has been widely used by CVD epidemiologists and researchers, and is prospectively associated with CVD and other health outcomes [7].

In 2022, the AHA released an updated definition of CVH called Life's Essential 8 (LE8) [7]. This definition includes all health behaviors and factors (diet, physical activity, nicotine exposure, body mass index (BMI), blood lipids, blood pressure, and blood glucose) from LS7, and adds sleep as a new component [7]. Additionally, in comparison to the LS7, the LE8 updates the scoring of the 8 metrics to be continuous to better represent interindividual differences, and changes the definitions of component criteria to be in line with current clinical guidelines [7].

Although associations of social isolation, social support, trait anger, and depressive symptoms with individual components of CVH are relatively well studied [4,8], fewer prior studies have focused on the associations between these factors and the AHA's CVH composite score. The majority of existing studies examined associations between depressive symptoms and CVH, while very few investigated associations of social isolation, social support, or trait anger with CVH [9–18]. To our knowledge, no studies have investigated the associations of social isolation, social support, trait anger, and depressive symptoms with CVH as defined by the new LE8 score. Examination of the associations of social isolation, social support, trait anger, and depressive symptoms with CVH may suggest targets to improve CVH before CVD manifests, or may indicate individuals in need of cardiovascular (CV) risk factor intervention based on their psychosocial factor profile.

Therefore, we used data from the Atherosclerosis Risk in Communities (ARIC) Study to investigate the cross-sectional associations of these four psychosocial factors with CVH defined by the AHA's LE8, and with individual LE8 components.

## Methods

### Study design and population

The ARIC study is an ongoing community-based, prospective cohort study of middle-aged mostly White and Black men and women. Details of the ARIC study have been previously published [19]. Briefly, from 1987–1989, 15,792 men and women ages 45–64 years were recruited from four communities in the United States (Forsyth County, NC; Jackson, MS; suburbs of Minneapolis, MN; and Washington County, MD) using various probability sampling methods described in detail elsewhere [20]. The baseline study visit (Visit 1, 1987–1989) was followed by additional in-person evaluations, along with yearly follow-up telephone calls and semi-annual follow-up calls starting in 2012. Protocols for all in-person visits were approved by the institutional review boards at the University of North Carolina at Chapel Hill, Wake Forest University, University of Mississippi Medical Center, University of Minnesota, and Johns Hopkins University. All participants provided written informed consent. Data were originally accessed for this analysis on September 11, 2022 and are can be requested through the Biologic Specimen and Data Repository Information Coordinating Center (BioLINCC) website (https://biolincc.nhlbi.nih.gov/studies/aric/) after registering with the site.

Of the 14,348 ARIC cohort members who participated in Visit 2 (1990–1992), we excluded those with self-reported race other than Black or White (n = 42) and Black participants from Washington County, MD or the suburbs of Minneapolis, MN (n = 49) due to small samples sizes. We also excluded participants with missing social isolation, social support, trait anger, and depressive symptom data (n = 1,363), and those missing a component of CVH: diet (n = 256), physical activity (n = 4), nicotine exposure (n = 4), blood lipid measures (n = 132), blood pressure (n = 48), blood glucose measures (n = 54), BMI (n = 0), and sleep measures (n = 0). Those with prevalent coronary heart disease, myocardial infarction, and stroke, at Visit 2 were further excluded (n = 1,038), along with participants who were missing data on covariates of interest (n = 16). Our final analytic sample for analyses included 11,311 ARIC participants.

### Psychosocial factors: Social isolation, social support, trait anger, and depressive symptoms

The four psychosocial factors were measured using self-administered questionnaires at ARIC Visit 2 (1990–1992). Social isolation addresses the number and frequency of interactions with social contacts, along with involvement in social networks and the community [21]. Social isolation was assessed using the Lubben Social Network Scale, a psychometrically valid questionnaire that has high internal reliability [22]. This questionnaire includes 10 questions on the self-assessed availability of social interactions that use a 0–5 rating scale with total scores ranging from 0–50. Previous studies have categorized this measure into levels of risk for social isolation: low risk (31–50), moderate risk (26–30), high risk (21–25), and socially isolated (0–20) [23]. Perceived social support, the extent and types of support available from existing relationships [24], was measured using an abbreviated, 16-item version of the Interpersonal Support Evaluation List. This questionnaire has good internal consistency, and is highly correlated with other measures of social support [25,26]. Total scores ranged from 0–48.

Anger, an emotion that arises from feelings of being treated unfairly and is accompanied by an agitated state [4], was measured using the Spielberger Trait Anger Scale. This questionnaire measures trait anger, and includes 10 questions that measure the frequency and intensity of symptoms of anger, particularly concepts such as hostility, and anger [27]. Response options are scored from 1–4, with total scores ranging from 10–40; previous studies have categorized

this measure as low trait anger (10–14), moderate trait anger (15–21), and high trait anger (22–40) [28]. Depressive symptoms were measured using the Maastricht Vital Exhaustion Questionnaire [29]. We utilized 18 of the 21 questions as a measure of depressive symptoms, excluding three questions most related to sleep quality. Questions were scored as 0 (No), 1 (Don't Know), and 2 (Yes), and overall summary scores ranged from 0–36 with higher scores indicating greater number of depressive symptoms.

## Cardiovascular health

CVH was defined using the AHA's new definition, the LE8 [7]. Measurement of the 8 CVH metrics (diet [30], physical activity [31], nicotine exposure, sleep, BMI, blood lipids, blood pressure, and blood glucose [32]), scoring details, and any deviations from LE8 definitions due to data availability are outlined in Table 1 (diet data summarized in S1 Table). Briefly, each LE8 component was calculated on a scale of 0–100. A measure of hours of sleep was not available; therefore, we used a derived measure of sleep quality from the Maastricht Vital Exhaustion Questionnaire. Total LE8 was calculated as the average of all LE8 health metrics, and ranged from 0–100 with higher scores indicating better CVH(7). The total summed LE8 score was categorized per AHA guidance as high (80≤ to ≤100), moderate (50≤ to <80), and low (0≤ to <50) CVH [7]. All LE8 components were measured at Visit 2, except diet and physical activity, which were measured at Visit 1 and are used as an approximation of these measures at Visit 2.

## Covariates

Biologic sex (male or female), self-reported race (Black or White), age (years), and education (< 12 years, 12 years, 12 < to < 16 years, and ≥ 16 years) were assessed at ARIC Visit 1. Age at Visit 1 was used to calculate age at Visit 2 by adding the time between visits to the participant's age at Visit 1, rounding down to a whole number. Due to the sampling strategies, race and study-center are highly correlated in the ARIC cohort. Therefore, the variable race-center was developed as a 5-level categorical classification of race and field center (MN/White, MD/White, NC/White, NC/Black, MS/Black). Covariates for inclusion in models were identified based on prior literature and a priori scientific knowledge.

## Statistical analysis

Tertiles of high, moderate, and low social support and depressive symptoms were computed, as there are no established categories for these measures. We produced descriptive statistics of demographic characteristics, psychosocial variables, and LE8 components in the overall analysis sample and stratified by levels of LE8 (high, moderate, low). Additionally, the prevalence of high, moderate, and low levels of LE8 were computed by high, moderate, and low levels of psychosocial factors. The cross-sectional associations between individual psychosocial factors with continuous overall LE8 score and continuous LE8 component scores were examined using multivariable linear regressions adjusted for sex, race-center, age, and education level. These covariates were identified a priori using substantive knowledge and prior literature. We assessed effect measure modification of these associations by sex and race by including interaction terms with psychosocial exposures. If p-values (2-sided) for interaction terms were < 0.05, models were stratified by modifying factors.

Although hours of sleep was not measured at Visit 2, self-reported typical hours of sleep per night was measured on a subset of 1,604 ARIC participants as a part of the Sleep Heart Health Study about 6 years (1996–1998) after Visit 2 [33,34]. Therefore, we conducted a sensitivity analysis using hours of sleep, as recommended by the LE8 scoring guidelines, instead of the

**Table 1. Scoring for Life's Essential 8 according to Llyod-Jones et al. 2022 [7], measurement of Life's Essential 8 metrics in the Atherosclerosis Risk in Communities (ARIC) Study, and modifications made to comply with data availability.**

| LE8 metric and definition | Measurement in ARIC | Score modifications for ARIC data |
|---|---|---|
| **Diet**: Population scoring using quantiles of DASH-style adherence or Healthy Eating Index:<br>$100 = \geq 95^{th}$ percentile (top/ideal)<br>$80 = 75^{th}$-$94^{th}$ percentile<br>$50 = 50^{th}$-$74^{th}$ percentile<br>$25 = 25^{th}$-$49^{th}$ percentile<br>$0 = 1^{st}$-$24^{th}$ percentile (bottom/least ideal) | A modified version of the 66-item Harvard food frequency questionnaire, which asks participants how frequently they eat a certain quantity of a specific food (eg. ½ cup serving of ice cream), with response options including: > 6 per day, 4–6 per day, 2–3 per day, 1 per day, 5–6 per week, 2–4 per week, 1 per week, 1–3 per month, and almost never.[30] | See S1 Table |
| **Physical activity**: Minutes of moderate-to-vigorous activity per week:<br>$100 = \geq 150$ mins<br>$90 = 120$–$149$ mins<br>$80 = 90$–$119$ mins<br>$60 = 60$–$89$ mins<br>$40 = 30$–$59$ mins<br>$20 = 1$–$29$ mins<br>$0 = 0$ mins | Measured by the Baecke questionnaire, which assesses the participant's yearly frequency of participating in sports (up to four entries) and their walking habits.[31] These logs were converted to metabolic equivalents (METs) per the Compendium of Physical Activities, and weekly minutes of moderate-to-vigorous physical activity was calculated by multiplying the duration of weekly physical activity by the number of months per year. | Same |
| **Nicotine exposure**: Combustible tobacco use or nicotine-delivery system, or secondhand smoke exposure:<br>$100 = $ never smoker<br>$75 = $ former smoker, quit $\geq 5$ years<br>$50 = $ former smoker, quit 1–5 years<br>$25 = $ former smoker, quit < 1 year or currently using nicotine delivery system<br>$0 = $ current smoker | Smoking status was measured at Visits 1 and 2 as current, former, or never smokers. At Visit 2, participants were asked for how long they had stopped smoking. Additionally, non-smokers were asked how many hours/week they were exposed to second-hand smoke at Visit 2. | Cigarette smoking status:<br>$100 = $ never smoker<br>$75 = $ former smoker, quit $\geq 3$ years, or non-smoker in the upper quartile of hours/week of exposure to second-hand smoke<br>$50 = $ former smoker, quit 1–3 years<br>$25 = $ former smoker quit < 1 year<br>$0 = $ current smoker |
| **Sleep**: Average hours of sleep per night:<br>$100 = 7$ to $\leq 9$ hours<br>$90 = 9$ to $\leq 10$ hours<br>$70 = 6$ to $\leq 7$ hours<br>$40 = 5$ to $\leq 6$ or > 10 hours<br>$20 = 4$ to $\leq 5$ hours<br>$0 = \leq 4$ hours | Sleep quality was assessed using the first three questions of the Maastricht Vital Exhaustion Questionnaire:<br>1) Do you often feel tired?,<br>2) Do you often have trouble falling asleep?,<br>3) Do you wake up repeatedly during the night?<br>Responses were scored on a scale of 0–2, with total scores ranging from 0–6 and higher scores indicating poorer sleep quality. | Score on Maastricht Vital Exhaustion sleep-related questions:<br>$100 = 0$<br>$90 = 1$<br>$70 = 2$<br>$60 = 3$<br>$40 = 4$<br>$20 = 5$<br>$0 = 6$ |
| **Body mass index**: BMI ($kg/m^2$):<br>$100 = < 25$ $kg/m^2$<br>$70 = 25.0$–$29.9$ $kg/m^2$<br>$30 = 30.0$–$34.9$ $kg/m^2$<br>$15 = 35.0$–$39.9$ $kg/m^2$<br>$0 = \geq 40.0$ $kg/m^2$ | BMI was calculated as weight (kg) / height squared ($m^2$), where weight was measured using a balance beam scale, and height was measured in cm by a stadiometer. | Same |
| **Blood lipids**: Plasma total cholesterol and HDL cholesterol used to calculate non-HDL cholesterol (mg/dL):<br>$100 = < 130$ mg/dL<br>$60 = 130$–$159$ mg/dL<br>$40 = 160$–$189$ mg/dL<br>$20 = 190$–$219$ mg/dL<br>$0 = \geq 220$ mg/dL<br>If drug-treated level, subtract 20 points | Total cholesterol, high-density lipoproteins, and low-density lipoproteins were measured using standardized enzymatic methods on fasted blood samples. Cholesterol-lowering medication use within the past two weeks were taken by self-report and or from prescription bottles. | Same |
| **Blood glucose**: Fasting Blood Glucose (FBG) mg/dL or HbA1c (%):<br>$100 = $ No history of diabetes and FGB < 100 (or HbA1c < 5.7)<br>$60 = $ No diabetes and FBG 100–125 (or HbA1c 5.7–6.4)<br>$40 = $ Diabetes with HbA1c < 7.0<br>$30 = $ Diabetes with HbA1c 7.0–7.9<br>$20 = $ Diabetes with HbA1c 8.0–8.9<br>$10 = $ Diabetes with HbA1c 9.0–9.9<br>$0 = $ Diabetes with HbA1c $\geq 10.0$ | Serum blood glucose and Hemoglobin A1c (HbA1c) were both measured as part of fasted blood samples at ARIC Visit 2, with HbA1c measured retrospectively as part of an ancillary study. Serum blood glucose levels were measured using the hexokinase/glucose-6-phosphate dehydrogenase procedure. HbA1c was measured using the Tosoh 2.2 Plus HPLC instrument and the Tosoh G7 HPLC instrument.[32] Intraclass correlation between these two instruments was 0.98.[32] | Same<br>Diabetes defined as fasting blood glucose $\geq 126$ mg/dL |

*(Continued)*

**Table 1.** (Continued)

| LE8 metric and definition | Measurement in ARIC | Score modifications for ARIC data |
|---|---|---|
| **Blood pressure**: Systolic/diastolic blood pressures (mmHg)<br>100 = <120 / < 80 mmHg<br>75 = 120–129 / < 80 mmHg<br>50 = 130–139 or 80–89 mmHg<br>25 = 140–159 or 90–99 mmHg<br>0 = ≥ 160 or ≥ 100 mmHg<br>Subtract 20 points if treated level | Sitting blood pressures (systolic and diastolic, mmHg) were measured 3 times after a 5 minute rest period, using a random zero sphygmomanometer; the final two measurements were averaged and used in analyses. Antihypertensive use within the past two weeks was taken by self-report and from prescription bottles. | Same |

modified sleep quality measure used for primary analyses among this sample. The correlation between hours of sleep per night and sleep quality-related questions from the Vital Exhaustion Questionnaire was assessed among the participants who have both measures. All analyses were repeated among this subsample of participants using hours of sleep to derive LE8. Another sensitivity analysis was conducted adjusting depressive symptom analyses for depression-related medications measured at Visit 2. As a further sensitivity analyses, we assessed the same associations of psychosocial factors with the odds of having a high LE8 score compared to moderate/low LE8 score using adjusted logistic regressions. All analyses were conducted in SAS version 9.4 (SAS Institute, Cary, NC, United States).

## Results

### Participant characteristics

Participants had a mean age of 56.8 (standard deviation (SD): 5.7) years, were 58% female, and 23% Black. Nineteen percent had less than a high school education, and the mean (SD) BMI was 27.9 (5.4) kg/m$^2$ (Table 2). The cohort consisted of 22% current smokers and 13% participants with diabetes. Mean (SD) minutes of weekly moderate-to-vigorous physical activity was 135 (154) minutes/week, while mean (SD) systolic blood pressure was 121 (19) mmHg. One percent were considered socially isolated, with another 4% at high-risk for social isolation. The prevalence of high trait anger was 7%. Mean (SD) social support score was fairly high at 37.5 (6.2) with a range of 4 to 48. Mean (SD) depressive symptoms score was relatively low at 7.9 (7.1) with a range of 0 to 36.

Compared to those with scores in the low LE8 category, the high LE8 category included more females, White participants, and those with higher education levels (Table 2). Additionally, those with high LE8 were less likely to be socially isolated/at high-risk of social isolation, less likely to have high trait anger, and had higher mean social support scores and lower mean depressive symptom scores than those in the moderate or low LE8 categories. The prevalence of high, moderate, and low overall LE8 levels also varied by levels of psychosocial factors, with high LE8 being more prevalent among low social isolation, low trait anger, low depressive symptoms, and high social support, compared to less favorable levels of the four psychosocial factors (Fig 1).

### Psychosocial factors and cardiovascular health

Moderate and high levels of social isolation, trait anger, and depressive symptoms were incrementally associated with lower LE8 scores compared to low levels of these psychosocial factors (Fig 2). Similarly, low and moderate levels of social support, compared to high levels of social support, were incrementally associated with lower LE8 scores. The association between social isolation and LE8 was the smallest in magnitude, while depressive symptoms had the

**Table 2. Sociodemographic, psychosocial characteristics, and health-related characteristics of the analysis sample at ARIC Visit 2 (1990–1992), stratified by Life's Essential 8 cardiovascular health category; N = 11,311.**

| | N (%) or mean ± standard deviation | | |
|---|---|---|---|
| | | Level of cardiovascular health | |
| | All participants | Low (0–49) | Moderate (50–79) | High (80–100) |
|---|---|---|---|---|
| Total | 11,311 | 2491 (22.0) | 7530 (66.6) | 1290 (11.4) |
| Study center | | | | |
| Forsyth County, NC | 3081 (27.2) | 507 (20.4) | 2134 (28.3) | 440 (34.1) |
| Jackson, MS | 2310 (20.4) | 931 (37.4) | 1306 (17.3) | 73 (5.7) |
| Suburbs of Minneapolis, MN | 3239 (28.6) | 485 (19.5) | 2262 (30.0) | 492 (38.1) |
| Washington County, MD | 2681 (23.7) | 568 (22.8) | 1828 (24.3) | 285 (22.1) |
| Female sex | 6509 (57.6) | 1475 (59.2) | 4145 (55.1) | 8889 (68.9) |
| Black (race) | 2598 (23.0) | 1027 (41.2) | 1485 (19.7) | 86 (6.7) |
| Age at Visit 2 | 56.8 ± 5.7 | 56.8 ± 5.6 | 56.9 ± 5.7 | 56.1 ± 5.6 |
| Education | | | | |
| < 12 years | 2164 (19.1) | 862 (34.6) | 1233 (16.4) | 69 (5.4) |
| 12 years | 3570 (31.6) | 796 (32.0) | 2445 (32.5) | 39 (25.5) |
| >12 years to < 16 years | 991 (8.8) | 205 (8.2) | 685 (9.1) | 101 (7.8) |
| ≥ 16 years | 4586 (40.5) | 628 (25.2) | 3167 (42.1) | 791 (61.3) |
| Social isolation | | | | |
| Socially isolated (≤20) | 140 (1.2) | 46 (1.9) | 82 (1.1) | 12 (0.9) |
| High risk (21–25) | 459 (4.1) | 131 (5.3) | 287 (3.8) | 41 (3.2) |
| Moderate risk (26–30) | 1471 (13.0) | 377 (15.1) | 945 (12.6) | 149 (11.6) |
| Low risk (≥31) | 9241 (81.7) | 1937 (77.8) | 6216 (82.6) | 1088 (84.3) |
| Social support | 37.5 ± 6.2 | 35.9 ± 6.7 | 37.7 ± 5.9 | 39.0 ± 5.5 |
| Trait anger | | | | |
| High (22–40) | 787 (7.0) | 265 (10.6) | 476 (6.3) | 46 (3.6) |
| Moderate (15–21) | 6268 (55.4) | 1450 (58.2) | 4166 (55.3) | 652 (50.5) |
| Low (10–14) | 4256 (37.6) | 776 (31.2) | 2888 (38.4) | 592 (45.9) |
| Depressive symptoms | 7.9 ± 7.1 | 11.2 ± 8.1 | 7.2 ± 6.6 | 5.1 ± 5.2 |
| Diet score | 41.4 ± 32.1 | 23.4 ± 26.8 | 43.6 ± 32.2 | 72.5 ± 25.4 |
| Minutes of weekly MVPA | 135 ± 154 | 36 ± 88 | 147 ± 153 | 254 ± 150 |
| Smoking status | | | | |
| Never smoker | 4635 (41.0) | 713 (28.6) | 3120 (41.4) | 802 (62.2) |
| Former smoker | 4233 (37.4) | 772 (31.0) | 3002 (39.9) | 459 (35.6) |
| Current smoker | 2443 (21.6) | 1006 (40.4) | 1408 (18.7) | 29 (2.3) |
| Sleep score | 69.5 ± 31.7 | 51.1 ± 35.0 | 72.7 ± 29.4 | 86.0 ± 20.5 |
| Body mass index | 27.9 ± 5.4 | 31.4 ± 6.3 | 27.4 ± 4.7 | 23.9 ± 2.7 |
| Non-HDL cholesterol (mg/dL) | 159 ± 42 | 178 ± 43 | 158 ± 40 | 131 ± 32 |
| Diabetes | 1490 (13.2) | 756 (30.5) | 714 (9.5) | 20 (1.6) |
| HbA1c | 5.7 ± 1.1 | 6.3 ± 1.7 | 5.6 ± 0.9 | 5.3 ± 0.4 |
| Systolic blood pressure | 121 ± 19 | 131 ± 20 | 120 ± 17 | 110 ± 12 |
| Diastolic blood pressure | 72 ± 10 | 76 ± 11 | 72 ± 10 | 67 ± 8 |

N: Number; SD: Standard deviation; NC: North Carolina; MS: Mississippi; MN: Minnesota; MD: Maryland; MVPA: Moderate-to-vigorous physical activity; HDL: High-density lipoprotein; HbA1c: Hemoglobin A1c.

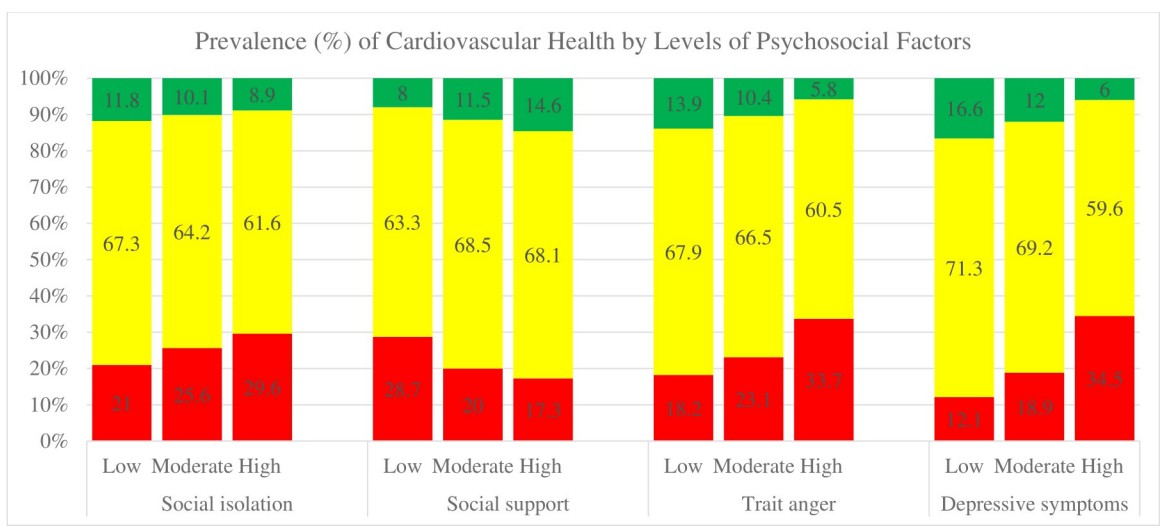

Low cardiovascular health = 0-49; moderate cardiovascular health = 50-79; high cardiovascular health = 80-100
Social isolation: low risk ≥ 31,  25 < moderate risk ≤ 30,  high risk/socially isolated  ≤ 25
Social support: 4 ≤ low < 36,  36 ≤ moderate < 41,  41 ≤ high ≤ 48
Trait anger: 10 ≤ low < 15,  15 ≤ moderate < 22,  22 ≤ high ≤ 40
Depressive symptoms: 0 ≤ low < 4,  4 ≤ moderate < 10,  10 ≤ high ≤36

**Fig 1. Prevalence of cardiovascular health (low/red, moderate/yellow, and high/green) by high, moderate, and low levels of psychosocial factors; N = 11,311.**

association of the greatest magnitude: being in the highest tertile of depressive symptoms was associated with having an LE8 score about 8 points lower than being in the lowest tertile of depressive symptoms (S2 Table). The associations between all four psychosocial factors and CVH were not modified by sex or race (data not shown).

When we explored individual components of LE8, high social isolation was associated with poor LE8 scores for diet, physical activity, nicotine use, and sleep quality, but was associated with a more favorable LE8 score for BMI (Table 3). Low social support, high trait anger, and high depressive symptoms were similarly associated with poor scores for diet, physical activity, nicotine use, and sleep quality. High levels of trait anger and depressive symptoms were associated with poor scores for BMI, and only the highest level of depressive symptoms was associated with having a poor score for blood glucose. None of the four psychosocial factors were associated with LE8 scores for blood lipids or blood pressure.

## Sensitivity analyses

Mean (SD) self-reported typical number of hours of sleep per night at approximately 6 years after Visit 2 was 7.3 (1.1) hours, with a range of 2.0 to 11.4 hours among the ARIC subsample that participated in the Sleep Heart Health Study (n = 1,604). The LE8 sleep score at Visit 2, defined using quality of sleep, showed a low degree of correlation with both duration of sleep (hours/night), and the sleep duration-based LE8 sleep score (Spearman correlation coefficients 0.12 and 0.14 respectively). Associations of psychosocial factors with overall LE8 score were closer to the null in this subsample (S3 Table). When overall LE8 scores were calculated using sleep duration instead of sleep quality, the magnitudes of association between psychosocial factors and LE8 were of an even smaller magnitude. The associations of each psychosocial factor with the LE8 sleep component specific score based on sleep duration were also closer to the null compared to those defined using sleep quality measures.

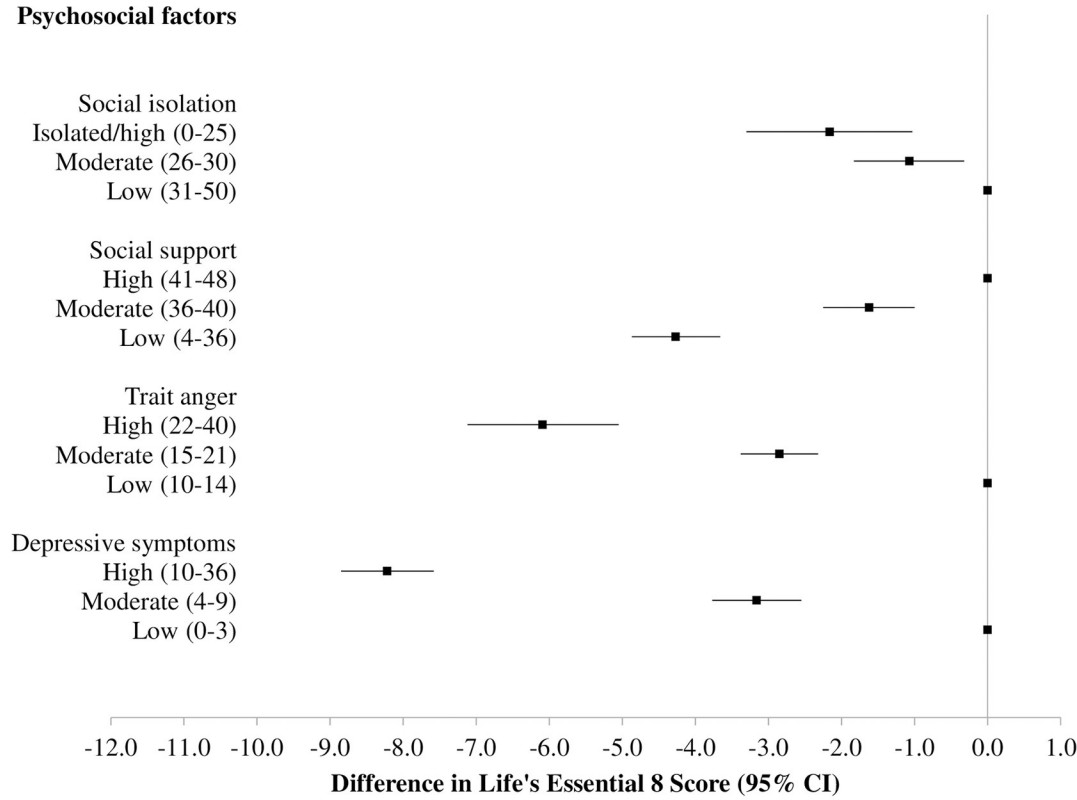

Models adjusted for sex, race-center, age, and education.
Estimates and 95% confidence intervals shown in Supplemental Table 2

**Fig 2. Cross-sectional adjusted associations of psychosocial factors with Life's Essential 8 using multivariable linear regressions, estimates represented as squares with 95% confidence interval bars; N = 11,311.**

In another sensitivity analysis, there were 439 participants (3.9%) taking depression-related medications at Visit 2, and further adjustment for these medications in depressive symptom models did not alter estimates or inferences (data not shown). When assessing the odds of high LE8 scores, the presence and direction of associations agreed with analyses of LE8 modeled continuously.

## Discussion

In cross-sectional analyses, social isolation, social support, trait anger, and depressive symptoms were modestly associated with having a 2 to 8 point lower CVH health score defined by AHA's LE8. When comparing the least to most favorable levels of these psychosocial factors, LE8 scores were 2 to 8 points lower. The associations between the four psychosocial factors and LE8 score did not differ by sex or race. Overall, the prevalence of poor psychosocial factors in this sample was fairly low.

Social isolation, social support, trait anger, and depressive symptoms may impact biological mechanisms related to the immune, neuroendocrine, and CV system via the following proposed pathways [4,5,7,8,35–37]. Negative psychosocial factors, such as high levels of anger, depressive symptoms, social isolation, or lack of social support, may promote poor coping health behaviors, increase stress, and decrease tangible social support resources [4,5,8,24]. Furthermore, depressive symptoms and distress caused by social isolation or lack of social support

**Table 3. Cross-sectional associations of psychosocial factors with cardiovascular health as defined by the American Heart Association's Life's Essential 8 definition at midlife, using multivariable linear regressions; The ARIC Study (1990–1992), N = 11,311.**

| | β (95% Confidence Interval) | | | | | | | |
|---|---|---|---|---|---|---|---|---|
| | **Diet** | **MVPA** | **Nicotine** | **Sleep** | **BMI** | **Lipids** | **Glucose** | **BP** |
| **Social isolation** | | | | | | | | |
| Isolated/high | -2.62 (-5.19, -0.06) | -6.53 (-10.07, -2.98) | -11.17 (-14.18, -8.16) | -6.76 (-9.29, -4.23) | 5.15 (2.72, 7.57) | 2.28 (-0.27, 4.83) | 0.06 (-1.99, 2.10) | 2.30 (-0.33, 4.92) |
| Moderate | 0.23 (-1.48, 1.94) | -2.88 (-5.24, -0.52) | -6.64 (-8.65, -4.64) | -2.99 (-4.68, -1.31) | 1.60 (-0.19, 3.22) | 0.96 (-0.74, 2.66) | 1.12 (-0.24, 2.48) | 0.01 (-1.74, 1.76) |
| Low | Referent | Referent | Referent | Referent | Referent | Referent | Referent | Referent |
| **Social support** | | | | | | | | |
| High | Referent | Referent | Referent | Referent | Referent | Referent | Referent | Referent |
| Moderate | -1.37 (-2.78, 0.05) | -4.92 (-6.87, -2.96) | -0.44 (-2.11, 1.22) | -5.80 (-7.18, -4.42) | -1.33 (-2.67, 0.02) | -0.08 (-1.49, 1.33) | 0.89 (-0.25, 2.02) | 0.06 (-1.39, 1.52) |
| Low | -4.50 (-5.87, -3.13) | -8.99 (-10.89, -7.10) | -5.89 (-7.50, -4.27) | -13.25 (-14.58, -11.91) | -1.17 (-2.47, 0.13) | 0.46 (-0.91, 1.83) | -0.62 (-1.71, 0.48) | -0.18 (-1.59, 1.23) |
| **Trait anger** | | | | | | | | |
| High | -7.07 (-9.42, -4.71) | -3.91 (-7.17, -0.64) | -13.96 (-16.73, -11.19) | -17.04 (-19.34, -14.73) | -3.76 (-6.00, -1.53) | 0.07 (-2.29, 2.42) | -1.50 (-3.38, 0.39) | -1.51 (-3.93, 0.91) |
| Moderate | -4.37 (-5.58, -3.17) | -1.90 (-3.57, -0.22) | -6.83 (-8.25, -5.41) | -7.42 (-8.60, -6.24) | -1.58 (-2.73, -0.44) | -1.04 (-2.24, 0.16) | -0.23 (-1.20, 0.74) | 0.55 (-0.69, 1.79) |
| Low | Referent | Referent | Referent | Referent | Referent | Referent | Referent | Referent |
| **Depressive symptoms** | | | | | | | | |
| High | -6.32 (-7.79, -4.85) | -9.73 (-11.76, -7.70) | -7.80 (-9.53, -6.06) | -32.94 (-34.27, -31.62) | -4.49 (-5.89, -3.10) | -0.64 (-2.10, 0.83) | -2.45 (-3.64, -1.29) | -1.36 (-2.87, 0.15) |
| Moderate | -3.57 (-4.99, -2.15) | -4.15 (-6.10, -2.19) | -3.53 (-5.20, -1.86) | -11.82 (-13.10, -10.55) | -1.21 (-2.56, 0.13) | 0.30 (-1.11, 1.71) | -0.36 (-1.49, 0.77) | -0.93 (-2.38, 0.53) |
| Low | Referent | Referent | Referent | Referent | Referent | Referent | Referent | Referent |

LE8: Life's Essential 8 cardiovascular health metric; MVPA: Moderate-to-vigorous physical activity; BMI: Body mass index; BP: Blood pressure.

Linear regressions adjusted for sex, race-center, age, and education.

Social isolation: Low risk ≥ 31, 25 < moderate risk ≤ 30, high risk/socially isolated ≤ 25.

Social support: 4 ≤ low < 36, 36 ≤ moderate < 41, 41 ≤ high ≤ 48.

Trait anger: 10 ≤ low < 15, 15 ≤ moderate < 22, 22 ≤ high ≤ 40.

Depressive symptoms: 0 ≤ low < 4, 4 ≤ moderate < 10, 10 ≤ high ≤ 36.

may negatively impact sleep [38,39]. Peer influence from social support and social connectedness may also alter health behaviors such as physical activity participation, smoking, diet, alcohol intake, medication adherence, and other health seeking behaviors [4,5,8,36,37]. Lastly, social connections and social support may affect health through access to health information and resources [24,37]. Some of these mechanisms may be bidirectional, as negative psychosocial factors may increase the risk of physical health conditions such as CVD as well as mental health problems, which may exacerbate already poor psychosocial factors by limiting social participation and causing further distress, such as depression or anger [7,36].

Although there are many hypothesized pathways that connect psychosocial factors and CVH, few studies have investigated these associations with CVH defined by AHA's metric LS7 or LE8 metric. Our study found modest cross-sectional associations of social isolation and social support with LE8 score. Our findings are in agreement with the previous two studies that assessed these associations [11,18]. Although our results suggested an incremental inverse association between trait anger and CVH, the only prior study that examined a related measure, hostility, found no association with CVH [11]. Differing results may be due to the fact

that hostility is a different construct than trait anger, different scoring methods used to assess CVH (LS7 vs LE8), or different study populations. Overall, our study in conjunction with prior evidence suggests an association exists between social isolation, social support, trait anger, and CVH. This study was one of the first to investigate associations of social isolation, social support, and trait anger with individual components of CVH scores in a single population.

The results of this study add to the strong evidence for an association between greater depressive symptoms and lower CVH defined by the AHA LS7/LE8 metrics, as suggested by the majority of studies that examined depressive symptoms with LS7 cross-sectionally [9–16]. In addition to previous cross-sectional studies, three studies have investigated the prospective associations of depressive symptoms and CVH, finding depressive symptoms to predict worse LS7 scores prospectively and vice versa [40]. However, some of our findings differ from previous studies that investigated the association of depressive symptoms with individual CVH components, albeit previous studies used LS7 scoring while we used LE8. Based on the results of our study, prior CVH studies, and other epidemiological and mechanistic evidence, most evidence points to an association between depressive symptoms and CVH behavioral components such as diet [9,12,13], physical activity [9,10,12–14,16], and nicotine use [9,10,12–14,16,17,41]. Although this study did not find evidence of an association between depressive symptoms and CVH scores for BMI and blood glucose, multiple previous studies also suggest an association between depressive symptoms and CVH scores for BMI [9,12,14,16] and blood glucose [9,14,16,17], although these associations were typically smaller in magnitude than associations with the CVH behavioral factors. It is possible that we do not observe these associations due to the low prevalence of depressive symptoms in this cohort. Alternatively, differing results may be due to the use of a non-validated set of questions from the Maastricht Vital Exhaustion questionnaire to represent depressive symptoms in this study, as opposed to previous studies that have used more widely recognized measurement tools or a depression diagnosis.

## Strengths and limitations

To our knowledge, this is the first study to examine psychosocial factors with CVH defined by the new LE8 metric. The LE8 improves upon past scoring as it offers a greater number of categories for each CVH metric, allowing for greater interindividual variation and for participants to get credit for moderate levels of CVH [7]. Additionally, the LE8 score incorporates sleep, which has been consistently associated with CVD outcomes, and scoring for other metrics was updated to match the most current public health and clinical guidelines [7]. Although a few prior studies have investigated the associations between psychosocial factors and CVH defined by LS7, the current study was likely able to better characterize the dose response relationship between psychosocial factors and CVH as the LE8 allows for greater granularity of CVH due to its scoring on a semi-continuous scale of 0–100. However, as these were cross-sectional data, we cannot infer causal or temporal relationships between psychosocial factors and CVH from this data.

This study was limited in that it did not measure hours of sleep at Visit 2, and instead used a derived measure of sleep quality. Sleep duration and quality are conceptualized as separate components and are typically assessed as separate exposures [42]. While there is overlap between those who have particularly short or long sleep durations and those with poor sleep quality, both aspects of sleep measurement may be important to consider in terms of public health since sleep duration needs can vary by individual [42]. This may explain the weak correlation between LE8 scores based on sleep duration and LE8 scores based on sleep quality.

Additionally, the sleep quality questions may be inherently related to psychosocial factors since they were drawn from the measure of depressive symptoms, potentially inflating associations. Although the direction of associations of the four psychosocial factors with overall CVH score did not change when using LE8 based on sleep duration, further research is needed to determine the impact of defining LE8 sleep scores based on sleep duration or sleep quality. This study was also limited in that physical activity and diet were measured at Visit 1, approximately 3 years before Visit 2, which were used as an approximation of these measures at Visit 2. Lastly, self-reported measures such as diet, smoking status, and physical activity may be subject to recall and social desirability biases. This may increase measurement error and would likely cause results to be an underestimate of the true association due to potential underreporting of poor smoking, diet, or physical activity habits.

## Conclusions

Results from this study expand on the previously established associations of psychosocial factors with CVD prevalence and incidence by assessing how they are associated with CVH and its components before clinical CVD manifests. This evidence suggests that those with the poorest psychosocial risk factors, particularly people with high depressive symptoms, may be an important group to target for CVD risk factor intervention, as they have the poorest CVH scores.

## Supporting information

**S1 Table. Details of diet scoring, comparing the foods included on the Life's Essential 8 scoring by Lloyd-Jones et al.** 2022 [7] and the scoring used in the Atherosclerosis Risk in Communities (ARIC) study based on diet item availability.
(DOCX)

**S2 Table. Cross-sectional adjusted associations of psychosocial factors with cardiovascular health defined by the American Heart Association's Life's Essential 8 metric at ARIC Visit 2, using multivariable linear and logistic regressions; N = 11,311.**
(DOCX)

**S3 Table. Cross-sectional adjusted associations of psychosocial factors with cardiovascular health, defined by Life's Essential 8, and Life's Essential 8 sleep score, comparing scores defined using hours of sleep and sleep quality; N = 1,578.**
(DOCX)

## Acknowledgments

The authors thank the staff and participants of the ARIC study for their important contributions.

## Author Contributions

**Conceptualization:** Kennedy M. Peter-Marske, Anna Kucharska-Newton, Eugenia Wong, Yejin Mok, Priya Palta, Pamela L. Lutsey, Wayne Rosamond.

**Formal analysis:** Kennedy M. Peter-Marske.

**Investigation:** Kennedy M. Peter-Marske.

**Methodology:** Kennedy M. Peter-Marske, Anna Kucharska-Newton, Eugenia Wong, Yejin Mok, Priya Palta, Pamela L. Lutsey, Wayne Rosamond.

**Supervision:** Wayne Rosamond.

**Visualization:** Kennedy M. Peter-Marske.

**Writing – original draft:** Kennedy M. Peter-Marske.

**Writing – review & editing:** Kennedy M. Peter-Marske, Anna Kucharska-Newton, Eugenia Wong, Yejin Mok, Priya Palta, Pamela L. Lutsey, Wayne Rosamond.

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
