## [Decision Letter · Decision Letter 0]

1 Apr 2024

PONE-D-23-24863Associations of psychosocial factors and cardiovascular health measured by Life’s Essential 8: the Atherosclerosis Risk in Communities (ARIC) StudyPLOS ONE

Dear Dr. Peter-Marske,

Thank you for submitting your manuscript to PLOS ONE. After careful consideration, we feel that it has merit but does not fully meet PLOS ONE’s publication criteria as it currently stands. Therefore, we invite you to submit a revised version of the manuscript that addresses the points raised during the review process.

We look forward to receiving your revised manuscript.

Kind regards,

Mohammad Reza Mahmoodi, Ph.D.

Academic Editor

PLOS ONE

Journal Requirements:

Reviewers' comments:

Reviewer's Responses to Questions

**Comments to the Author**

1. Is the manuscript technically sound, and do the data support the conclusions?

Reviewer #1: Yes

Reviewer #2: Partly

2. Has the statistical analysis been performed appropriately and rigorously? 

Reviewer #1: Yes

Reviewer #2: Yes

3. Have the authors made all data underlying the findings in their manuscript fully available?

Reviewer #1: No

Reviewer #2: Yes

4. Is the manuscript presented in an intelligible fashion and written in standard English?

Reviewer #1: Yes

Reviewer #2: Yes

5. Review Comments to the Author

Reviewer #1: Abstract

In the methods section, it would be useful to mention the study setting. Indicate whether primary or secondary data source was used. Also, would be useful to mention the categories of scores used in the analysis.

Background

Some grammatical errors that need proof-reading. E.g., No full stop on line 80, In line 88 full stop should come after the citation and this should be revised throughout the background and other parts of the manuscript.

Sentence in line 81 that ends with ‘CVD events’ needs to be supported by a reference.

In line 104 CV is not expanded anywhere in preceding text. Please expand the abbreviation.

Methods

Please add some brief information on the sampling strategy for the ARIC study.

In the covariate section, please provide further details how visit 1 variables were used to derive visit 2 estimates i.e. for age. For the race-center variable, please provide some context why MN and MD don’t have black and MS doesn’t have white.

Did you consider using ordinal logistic regression as the levels of cardiovascular health seem to be ordinal in nature? If so, why wasn’t ordinal considered. It would be good to have this as a sensitivity analysis to support the linear regression results.

The figures seems to be a bit blur. Please provide a clearer version.

Results

For table 2 it would be useful to provide some indications of statistical differences from hypothesis tests e.g. chi-squared for categorical variables and t-test/ANOVA for continuous. You can use * to indicate significance against the variables and add as notes below the table.

Reviewer #2: Tile: OK

Abstract: I think it is essential to show the descriptive measures of Life’s Essential 8 scores in each group of psychosocial factor assessments. In this case, it is essential to show the classification of these scores in more detail in material and method part of this section. It seems the stated conclusion is not supported by the evidences already provided in the result part.

Keywords: OK

Introduction: OK

Material and method: please notice to the following issues in this section:

- Please show how ethical issues have been dealt with and who has approved these issues in the context of the paper.

- For a better understanding of the setting of the study, please define the inclusion and exclusion criteria of the recruited participants in more detail, I think referring to the paper of the original study is not enough for such a comprehensive paper.

- Since this is a cross sectional study, I think it is necessary to define the sampling method as well as the method of sample weighting (if necessary) in more detail.

- Please show the reliability measure of the questionnaire proposed for social isolation (line 138). This is true for the anger and depressive symptoms questionnaires as well (lines 150-158).

- Please define how the proposed variable of the study have been measured (in terms of time the method of invitation of the participants in to the study).

- I think it is essential to define how covariates were selected to be entered to the multivariate regression model.

Results: considering the following comments is appreciated in this section:

- Please show the P values of the comparisons of different variable in table 2. It also not necessary to express the findings already shown in the table once again in the context of the paper. Generally speaking, showing the univariate comparing analysis in the LE8 groups is mandatory before multivariate analysis.

- Please complete the legend of different figures (e.g., the colors of figure 1).

Discussion: OK

References: If possible, please use more recent references.

6. PLOS authors have the option to publish the peer review history of their article (what does this mean?). If published, this will include your full peer review and any attached files.

Reviewer #1: No

Reviewer #2: **Yes: **Babak Eshrati

---

## [Author Response · Author response to Decision Letter 0]

6 May 2024

Reviewer #1: Abstract

1. In the methods section, it would be useful to mention the study setting. Indicate whether primary or secondary data source was used. Also, would be useful to mention the categories of scores used in the analysis.

Thank you for this comment. We have added the following to the abstract:

“...in this secondary data analysis using cross-sectional data from the ARIC cohort study.”

The categories of LE8 scores were not used in multivariable analyses, so to clarify we have specified that we assessed continuous LE8 in the abstract methods section. We additionally added the categories of the psychosocial exposures (high, moderate, and low) to the methods section.

2. Background

Some grammatical errors that need proof-reading. E.g., No full stop on line 80,

The sentence referenced here ends at line 81, after listing a variety of cardiovascular-related diseases and their complications and progression.

3. In line 88 full stop should come after the citation and this should be revised throughout the background and other parts of the manuscript.

Thank you. This has been revised throughout the paper.

4. Sentence in line 81 that ends with ‘CVD events’ needs to be supported by a reference.

We thank the reviewer for this comment. References have been added for this sentence.

5. In line 104 CV is not expanded anywhere in preceding text. Please expand the abbreviation.

This has been fixed, thank you for noticing this error.

6. Methods

Please add some brief information on the sampling strategy for the ARIC study.

The ARIC study employed various sampling methods, of which have been previously described. It would take many sentences to fully describe this sampling, therefore we added a reference and statement to the methods reflecting this:

“...using various probability sampling methods described in detail elsewhere(19).”

7. In the covariate section, please provide further details how visit 1 variables were used to derive visit 2 estimates i.e. for age. 

The following text has been added: “by adding the time between visits to the participant’s age at Visit 1, rounding down to a whole number.”

8. For the race-center variable, please provide some context why MN and MD don’t have black and MS doesn’t have white.

This is due to the sampling strategy of the original ARIC cohort. The following text has been revised to reflect this and give a bit more context: “Due to the sampling strategies, race and study-center are highly correlated in the ARIC cohort. Therefore, the variable race-center was developed as a 5-level categorical classification of race and field center (MN/White, MD/White, NC/White, NC/Black, MS/Black).”

9. Did you consider using ordinal logistic regression as the levels of cardiovascular health seem to be ordinal in nature? If so, why wasn’t ordinal considered. It would be good to have this as a sensitivity analysis to support the linear regression results.

We ran all models as logistic regressions estimating the odds of having high LE8 scores compared to the odds of having moderate/low LE8, as has been commonly done when assessing LS7 scores. Associations were in the same direction as linear regression results. We have chosen not to include these models because one of the major benefits of the LE8 score compared to the LS7 is the granularity of the measure and its ability to give individuals more “credit” for intermediate measures of cardiovascular health on a scale of 0 to 100 rather than categorizing them simply into three bins. However, to the reviewer’s recommendation, we have added this as a sensitivity analysis in the methods and results section in case readers wonder about this.

“As a further sensitivity analyses, we assessed the same associations of psychosocial factors with the odds of having a high LE8 score compared to moderate/low LE8 score using adjusted logistic regressions.”

“When assessing the odds of high LE8 scores, the presence and direction of associations agreed with analyses of LE8 modeled continuously.”

10. The figures seems to be a bit blur. Please provide a clearer version.

We thank the reviewer for this comment. We also noticed that the figures became blurry when converted to the pdf to send to reviewers, but they were not burry in the versions that we submitted to the journal. We will work with the journal to ensure that the final copy of figures published is clear and easy to read.

11. Results

For table 2 it would be useful to provide some indications of statistical differences from hypothesis tests e.g. chi-squared for categorical variables and t-test/ANOVA for continuous. You can use * to indicate significance against the variables and add as notes below the table.

We thank the reviewer for this suggestion. We did not perform statistical tests for the variables listed in the descriptive Table 2 since these are descriptive statistics of the sample characteristics, and not presented for hypothesis testing. This method is suggested here:

“Inferential measures such as standard errors and confidence intervals should not be used to describe the variability of characteristics, and significance tests should be avoided in descriptive tables.” - https://www.acpjournals.org/doi/10.7326/0003-4819-147-8-200710160-00010-w1

12. Reviewer #2: Tile: OK

Abstract: I think it is essential to show the descriptive measures of Life’s Essential 8 scores in each group of psychosocial factor assessments. In this case, it is essential to show the classification of these scores in more detail in material and method part of this section. It seems the stated conclusion is not supported by the evidences already provided in the result part.

Due to word count limits (300 words total, currently at 293), we are not able to elaborate extensively on how psychosocial factors were assessed in the abstract. We have clarified that these measures were assessed categorically, and the scale direction of the LE8 score.

“Life’s Essential 8 was scored per the American Heart Association definition (0-100 range); higher scores indicate better cardiovascular health.”

“Associations of categories (high, moderate, and low) of each psychosocial factor”

We also changed the directionality of the language in the conclusion to better match that of the results section: “Less favorable measures of psychosocial health were associated with lower Life’s Essential 8 scores compared better measures of psychosocial health among middle-aged males and females.”

13. Keywords: OK

Introduction: OK

Material and method: please notice to the following issues in this section:

- Please show how ethical issues have been dealt with and who has approved these issues in the context of the paper.

Please see the following text in the methods section:

“Protocols for all in-person visits were approved by the institutional review boards at the University of North Carolina at Chapel Hill, Wake Forest University, University of Mississippi Medical Center, University of Minnesota, and Johns Hopkins University. All participants provided written informed consent.”

14. For a better understanding of the setting of the study, please define the inclusion and exclusion criteria of the recruited participants in more detail, I think referring to the paper of the original study is not enough for such a comprehensive paper.

The inclusion/exclusion criteria for the ARIC study are as follows and is as described in this manuscript: “men and women ages 45-64 years were recruited from four communities in the United States (Forsyth County, NC; Jackson, MS; suburbs of Minneapolis, MN; and Washington County, MD)”. 

There were no other major inclusion or exclusion criteria from the general ARIC cohort, please see the new reference 19 for confirmation of this. From this original cohort, we applied many exclusion criteria which are outlined in the second paragraph of the methods section.

15. Since this is a cross sectional study, I think it is necessary to define the sampling method as well as the method of sample weighting (if necessary) in more detail.

Sample weighting was not applicable to this study as this was a community-based cohort. The sampling methods for each study site differed slightly. Detailing the sampling method would involve much description of which has already been published; therefore reference 19 and the following sentence have been added:

“Briefly, from 1987-1989, 15,792 men and women ages 45-64 years were recruited from four communities in the United States (Forsyth County, NC; Jackson, MS; suburbs of Minneapolis, MN; and Washington County, MD) using various probability sampling methods described in detail elsewhere(19).”

16. Please show the reliability measure of the questionnaire proposed for social isolation (line 138). This is true for the anger and depressive symptoms questionnaires as well (lines 150-158).

The reliability of social isolation and trait anger have been previously published for different versions of these tools (using a different number of questions from the measures), therefore we did not report the reliability. Additionally, the internal consistency reported here for social support was for the 40-item measure, not the 16-item measure that is used in the present study, so this sentence has been revised to be more general:

“This questionnaire has good internal consistency, and is highly correlated with other measures of social support(24, 25).”

Depressive symptoms were derived using all but 3 questions from the Maastricht Vital Exhaustion questionnaire and therefore these 18 questions have not been validated as a measure of depressive symptoms. This was done so that we could incorporate the sleep-related items as a sleep measure in the LE8 score, and so that this measure would not be highly correlated with depressive symptoms since they are incorporated in the measure itself.

17. Please define how the proposed variable of the study have been measured (in terms of time the method of invitation of the participants in to the study).

Please see the following text outlining when participants were recruited and completed Visit 1: “The baseline study visit (Visit 1, 1987-1989) was followed by additional in-person evaluations, along with yearly follow-up telephone calls and semi-annual follow-up calls starting in 2012.”

The following text outlines how the psychosocial measures were administered and when: “The four psychosocial factors were measured using self-administered questionnaires at ARIC Visit 2 (1990-1992).”

For components of LE8 timing of measurement, please see the following sentence and Table 1 for details on their measurement: “All LE8 components were measured at Visit 2, except diet and physical activity, which were measured at Visit 1 and are used as an approximation of these measures at Visit 2.”

18. I think it is essential to define how covariates were selected to be entered to the multivariate regression model.

Thank you for this suggestion. The following sentence has been included in the methods section: “These covariates were identified a priori using substantive knowledge and prior literature.”

19. Results: considering the following comments is appreciated in this section:

- Please show the P values of the comparisons of different variable in table 2. It also not necessary to express the findings already shown in the table once again in the context of the paper. Generally speaking, showing the univariate comparing analysis in the LE8 groups is mandatory before multivariate analysis.

We thank the reviewer for this suggestion. We did not perform statistical tests for the variables listed in the descriptive Table 2 since these are descriptive statistics of the sample characteristics, and not presented for hypothesis testing. This method is suggested here:

“Inferential measures such as standard errors and confidence intervals should not be used to describe the variability of characteristics, and significance tests should be avoided in descriptive tables.” - https://www.acpjournals.org/doi/10.7326/0003-4819-147-8-200710160-00010-w1

20. Please complete the legend of different figures (e.g., the colors of figure 1).

Text for the figure legends includes the colors for figure 1:

“Fig 1. Prevalence of Cardiovascular Health (low/red, moderate/yellow, and high/green) by high/isolated, moderate, and low levels of psychosocial factors; N=11,311”

The following text has been added to clarify the figure legend for figure 2:

“Fig 2. Cross-sectional adjusted associations of psychosocial factors with Life’s Essential 8 using multivariable linear regressions, estimates represented as squares with 95% confidence interval bars; N=11,311”

21. Discussion: OK

References: If possible, please use more recent references.

Thank you for this comment. We have included Wulandari and Li as two more recent references of psychosocial factors and cardiovascular disease.

---

## [Decision Letter · Decision Letter 1]

5 Jun 2024

Associations of psychosocial factors and cardiovascular health measured by Life’s Essential 8: the Atherosclerosis Risk in Communities (ARIC) Study

PONE-D-23-24863R1

Dear Dr. Peter-Marske,

We’re pleased to inform you that your manuscript has been judged scientifically suitable for publication and will be formally accepted for publication once it meets all outstanding technical requirements.

Kind regards,

Mohammad Reza Mahmoodi, Ph.D.

Academic Editor

PLOS ONE

Additional Editor Comments (optional):

Reviewers' comments:

Reviewer's Responses to Questions

**Comments to the Author**

1. If the authors have adequately addressed your comments raised in a previous round of review and you feel that this manuscript is now acceptable for publication, you may indicate that here to bypass the “Comments to the Author” section, enter your conflict of interest statement in the “Confidential to Editor” section, and submit your "Accept" recommendation.

Reviewer #2: All comments have been addressed

2. Is the manuscript technically sound, and do the data support the conclusions?

Reviewer #2: Yes

3. Has the statistical analysis been performed appropriately and rigorously? 

Reviewer #2: Yes

4. Have the authors made all data underlying the findings in their manuscript fully available?

Reviewer #2: Yes

5. Is the manuscript presented in an intelligible fashion and written in standard English?

Reviewer #2: Yes

6. Review Comments to the Author

Reviewer #2: I think all of the comments are addressed by the distinguished authors and the paper can be accepted.

7. PLOS authors have the option to publish the peer review history of their article (what does this mean?). If published, this will include your full peer review and any attached files.

Reviewer #2: **Yes: **Babak Eshrati

---

## [Editor Report · Acceptance letter]

24 Jun 2024

PONE-D-23-24863R1 

PLOS ONE

Dear Dr. Peter-Marske, 

I'm pleased to inform you that your manuscript has been deemed suitable for publication in PLOS ONE. Congratulations! Your manuscript is now being handed over to our production team.

Kind regards, 

on behalf of

Dr. Mohammad Reza Mahmoodi 

Academic Editor

PLOS ONE